# SCENIR: Visual Semantic Clarity through Unsupervised Scene Graph Retrieval

**Nikolaos Chaidos** [1]   **Angeliki Dimitriou** [1]   **Maria Lymperaiou** [1]   **Giorgos Stamou** [1]

## Abstract

Despite the dominance of convolutional and transformer-based architectures in image-to-image retrieval, these models are prone to biases arising from low-level visual features, such as color. Recognizing the lack of semantic understanding as a key limitation, we propose a novel scene graph-based retrieval framework that emphasizes semantic content over superficial image characteristics. Prior approaches to scene graph retrieval predominantly rely on supervised Graph Neural Networks (GNNs), which require ground truth graph pairs driven from image captions. However, the inconsistency of caption-based supervision stemming from variable text encodings undermine retrieval reliability. To address these, we present *SCENIR*, a Graph Autoencoder-based unsupervised retrieval framework, which eliminates the dependence on labeled training data. Our model demonstrates superior performance across metrics and runtime efficiency, outperforming existing vision-based, multimodal, and supervised GNN approaches. We further advocate for *Graph Edit Distance* (GED) as a deterministic and robust ground truth measure for scene graph similarity, replacing the inconsistent caption-based alternatives for the first time in image-to-image retrieval evaluation. Finally, we validate the generalizability of our method by applying it to unannotated datasets via automated scene graph generation, while substantially contributing in advancing state-of-the-art in counterfactual image retrieval. The source code is available at https://github.com/nickhaidos/scenir-icml2025.

---

[1]Artificial Intelligence and Learning Systems Laboratory, National Technical University of Athens. Correspondence to: Nikolaos Chaidos <nchaidos@ails.ece.ntua.gr>.

*Proceedings of the 42$^{nd}$ International Conference on Machine Learning*, Vancouver, Canada. PMLR 267, 2025. Copyright 2025 by the author(s).

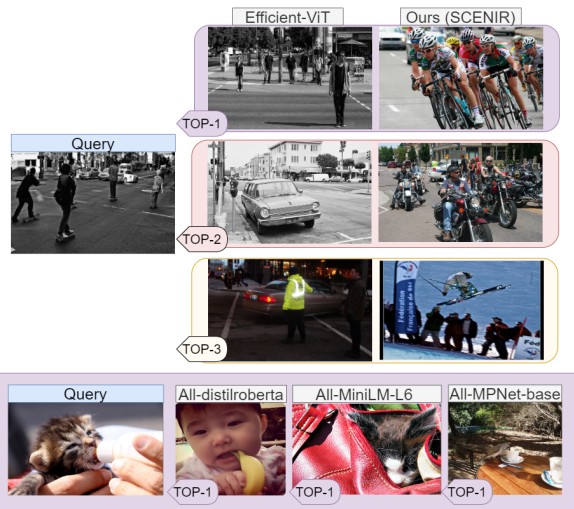

*Figure 1.* Top: Example of visual biases (color bias) in image retrieval when using visual models vs ours. Bottom: Example of the retrieval variability of SBERT models (top-1 retrieved items).

## 1. Introduction

With the advent of deep visual models, transiting from Convolutional Neural Networks (CNNs) to Vision Transformers (ViT), unprecedented performance has been reported across a variety of vision tasks. However, it is still unclear whether related models fully rely on correlational patterns or acquire an understanding of semantics, i.e. objects, attributes and their relationships. For example, CNNs lack understanding of contextual dependencies (Lin et al., 2020), while ViTs are widely exposed to correlation biases (Ghosal & Li, 2024), revealing that true knowledge of visual semantics has not yet been conquered (Wang et al., 2023b). To this end, there is an extensive line of work exposing erroneous results on state-of-the-art (SotA) visual models due to biases (Shi et al., 2022; Park & Kim, 2022; Wei et al., 2023; Murali et al., 2023; Menon et al., 2023; Izmailov et al., 2024; Puaduraru et al., 2024; Zheng et al., 2025).

We briefly demonstrate an example of biased image-to-image retrieval in Figure 1. Efficient-ViT (Cai et al., 2023) - a SotA image classifier - retrieves results that are predominantly based on color attributes, such as 'black & white'

rather than semantics (as seen in the top-1 and top-2 positions). This behavior strongly suggests that a disproportionate focus is placed on visual features like color. In turn, semantic relationships are overlooked - i.e the fact that the depicted group of people is riding a form of sports equipment (bikes, pairs of skis).

We argue that harnessing semantic information and mitigating visual biases can be tackled by adopting **scene graphs** in visual pipelines, as they provide structured representations of images, where the objects, attributes and relationships are explicitly presented (Chang et al., 2021). As a result, when a scene graph is employed for a vision task, such as image similarity -the focus of this work- *concepts* drive the decision-making of the underlying model. Thus, advanced interpretability and robustness is offered (Wang et al., 2023b), while redundant visual details

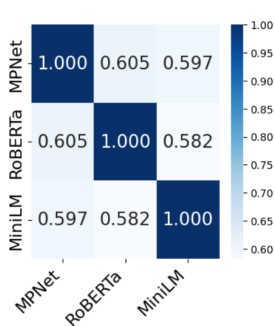

*Figure 2.* Agreement between top-1 retrieved items with various SBERT models (MPNet: all-mpnet-base-v2, RoBERTa: all-distilroberta-v1, MiniLM: all-minilm-l6-v2) for caption retrieval.

(lighting, camera angle, black & white image etc.) become less influential. Ultimately, the problem of image-to-image retrieval translates to that of **scene graph retrieval**.

However, this domain is not devoid of computational obstacles. Scene graph retrieval essentially requires solving graph matching and ultimately calculating Graph Edit Distance (GED) (Sanfeliu & Fu, 1983), an NP-hard problem (Zeng et al., 2009). To accelerate GED calculation, Graph Neural Networks (GNNs) have been utilized to acquire scene graph embeddings, facilitating efficient retrieval in lower-dimensionality spaces. Existing endeavors for GNN-based GED acceleration harness pre-calculated similarity scores between graph pairs, which demands obtaining ground truth GED scores beforehand (Dimitriou et al., 2024).

Staying within the supervised training spectrum, and by acknowledging the computational restrictions of GED calculation, other works resort to the utilization of captions as the ground truth supervision signals for GNN training (Yoon et al., 2021; Maheshwari et al., 2021; Wang et al., 2023b). Caption matching is based on pre-trained *Sentence-BERT (SBERT)* models (Reimers & Gurevych, 2019), which provide embedding representations of captions, allowing the computation of cosine similarity scores between their embeddings. Even this seemingly simple solution poses various disadvantages. To elaborate, there is *non-negligible variability* depending on the chosen SBERT model, result-

ing in different top-k matchings for a given query caption - as briefly demonstrated in Fig. 1. In general, SBERT models tend to return *disagreeing rankings*, when compared to each other; in Fig. 2, we calculate the agreement in top-1 retrieved results using some of the best-performing SBERT models for embedding captions[1]. As evidenced in the majority of results, there is **significant disagreement** ($\geq 40\%$) between the top-1 retrievals across models. Such inconsistencies in the supposed ground truth will inevitably cause *error cascading* in the trained GNN, ultimately leading to inconsistent outputs, as we demonstrate empirically in Section 4.2. On top of that, language as a ground truth modality should not be considered without apprehension. Captions are usually short descriptions, providing a very *high-level representation* of the image without comprehensive semantic details. Moreover, there is no definitive way to measure the similarity between two sentences, leading to dispute even among humans (Wang & Dong, 2020). All these issues suggest that deviating from captions as a ground truth GNN supervision signal is likely a more effective solution.

In this work, we propose a solution to bypass relying both on GED as well as image captions for GNN supervision. Instead, we favor **unsupervised** methods for scene graph retrieval, a topic that has been largely underexplored in previous research. Additionally, we highlight the need to scrutinize the evaluation strategies used for assessing scene graph retrieval systems, as they often lack reliable and robust ground-truth measures. To this end, we contribute to the following: ① We propose **SCENIR**, an *Unsupervised Graph Autoencoder* for scene graph retrieval that eliminates the need for similarity labels, while *surpassing supervised methods* in both performance and computational efficiency. ② We advocate the utilization of GED as the *standard evaluation framework* to compute reliable similarity scores without variability, therefore deterministically assessing the capacity of the models to capture important semantic information. ③ We present the *extendability* of our method on unannotated images via Scene Graph Generation (SGG) and offer use cases such as counterfactual image explanations.

## 2. Related work

**Graph Autoencoders** (GAEs) are at the center of our work. Despite the novel contribution of our proposed model, we utilize and build upon well-known GAE architectures, such as VGAE (Kipf & Welling, 2016) which learns graph representations by attempting to reproduce their original adjacency matrix from latent node embeddings in an encoder-decoder fashion. The Adversarially Regularized Variational Graph Autoencoder (ARVGA) (Pan et al., 2018) further

---

[1]Captions from 3000 images, from the PSG dataset(Yang et al., 2022) (under license CC-BY 4.0).

boosts the regularization of the latent embeddings via adversarial training. While both methods focus exclusively on topological aspects of the graph, the Graph Feature Autoencoder (GFA) (Hasibi & Michoel, 2021) implements a new decoder that solely regards node information, which learns to reconstruct the original feature matrix. In this paper, we propose a carefully designed combination and extension of architectural components to enhance the representative power of GAEs for semantic graphs.

**Graph similarity** methods leverage the topological structure and semantic content of graphs to measure their differences. While often a step in retrieval pipelines, their relevance here is complementary, as scene graph retrieval frameworks are a distinct field. Often employed in graph similarity endeavors, deterministic similarity/distance metrics such as GED (Sanfeliu & Fu, 1983) are computationally expensive, ultimately rendering the use of approximation frameworks like GNNs an imperative choice. Most widely used GNN models are supervised, i.e. they are trained using either a large amount of pre-computed similarity labels (Bai et al., 2018a;b; Zhuo & Tan, 2022), or ground-truth positive/negative pairs (Li et al., 2019; Ying et al., 2020). In contrast, SCENIR mitigates the extensive data requirements by employing unsupervised GAEs to encode scene graphs and then compute embedding similarity. GED is *only* used in the evaluation stage as a ground truth measure.

**Scene graph retrieval** though well-defined as a field, remains relatively underexplored in literature. In the general context of similarity, scene graphs have primarily been utilized for cross-modal text-image retrieval (Wang et al., 2019; Peng & Chi, 2020; Zhong et al., 2021; Wang et al., 2023a). However, despite its semantic advantages, *image-to-image retrieval through scene graphs*, the focus of our work, has received limited attention so far (Yoon et al., 2021; Maheshwari et al., 2021; Wang et al., 2023b). Yoon et al. (2021) (IRSGS) train a three-layer siamese GNN leveraging similarity labels derived from SBERT-embedded image captions. The same caption-based similarities are treated as ground truth for evaluation. Their probabilistic training method requires labels that scale quadratically with respect to the available scene graphs. Our approach differs from their framework in two major ways: a) relying entirely on unsupervised models for greater efficiency in training time and label requirements, and b) using the more robust GED as ground truth. Recent work by Wang et al. (2023b) merges visual and graph features for multi-modal image retrieval using images and scene graphs as input. Finally, the approach of Dimitriou et al. (2024) (Graph Counterfactuals - GC) explores GNN-based methods to produce counterfactual explanations (CEs) for scene classifiers, utilizing GED as the evaluation framework, similar to us. Although our primary focus is on scene graph retrieval rather than CEs,

we present a brief comparative use case within the counterfactual context. In doing so, we demonstrate that SCENIR is more competent in the inductive setting, a scenario not explored in their work.

## 3. Method

### 3.1. Notation & Problem Formulation

In this paper, we focus on image-to-image retrieval using scene graphs; thus, a suitable dataset comprises $(I, G)$ pairs, where $I$ is an image, and $G$ the corresponding scene graph. Formally, a scene graph $G = (V, E)$ consists of a set of nodes $V$ and a set of edges $E \subseteq V \times V$. It is specifically formulated as a *feature matrix* $\mathbf{X} \in \mathbb{R}^{n \times d}$ (information about objects in image $I$), and an *adjacency matrix* $\mathbf{A} \in \mathbb{R}^{n \times n}$ (information about relations in image $I$), where $n = |V|$ is the number of nodes, and $d$ the feature vector dimensionality per node. A scene graph retrieval framework receives as input the query pair $(I_q, G_q)$ and the candidate pairs $\{(I_{c_1}, G_{c_1}), (I_{c_2}, G_{c_2}), \ldots (I_{c_n}, G_{c_n})\}$. The output is a permutation of the candidates $\{(I_{k_1}, G_{k_1}), (I_{k_2}, G_{k_2}), \ldots (I_{k_n}, G_{k_n})\}$, which aims to approximate the ranking of a surrogate similarity measure $sim(G_q, G_{c_i})$ defined on the scene graphs. For this similarity measure we utilize GED, as proposed in Dimitriou et al. (2024), where, instead of exclusively utilizing it for evaluating counterfactual explanations, we extend it to evaluate scene graph retrieval in general.

Here, we harness **GNNs** to extract global embeddings for query and candidate scene graphs, and then calculate the final rankings through cosine similarity. Despite this retrieval pipeline being conceptually simple, using *unsupervised representations for scene graph retrieval* is surprisingly an underexplored topic. To this end, our experimentation revolves around GAE models, which we integrate to our retrieval framework as sub-modules to compute embeddings.

### 3.2. Proposed Model

While existing GAE approaches have shown promise in graph-based tasks, they face key limitations with scene graph retrieval, including struggles with complex node-edge representations, limited expressiveness of inner-product decoders, and loss of discriminative power due to oversmoothing. Our proposed **SCENIR** (*SCene-graph auto-ENcoder for Image Retrieval*) framework addresses these challenges through a carefully designed combination of split encoders, MLP-based decoders, and adversarial training.

**Encoder** The encoder comprises two GNN modules $(GNN_\mu$ and $GNN_\sigma)$, whose goal is to encode the input graph into a learned latent space. It takes as input the scene graph's feature matrix $\mathbf{X}$ and adjacency matrix $\mathbf{A}$,

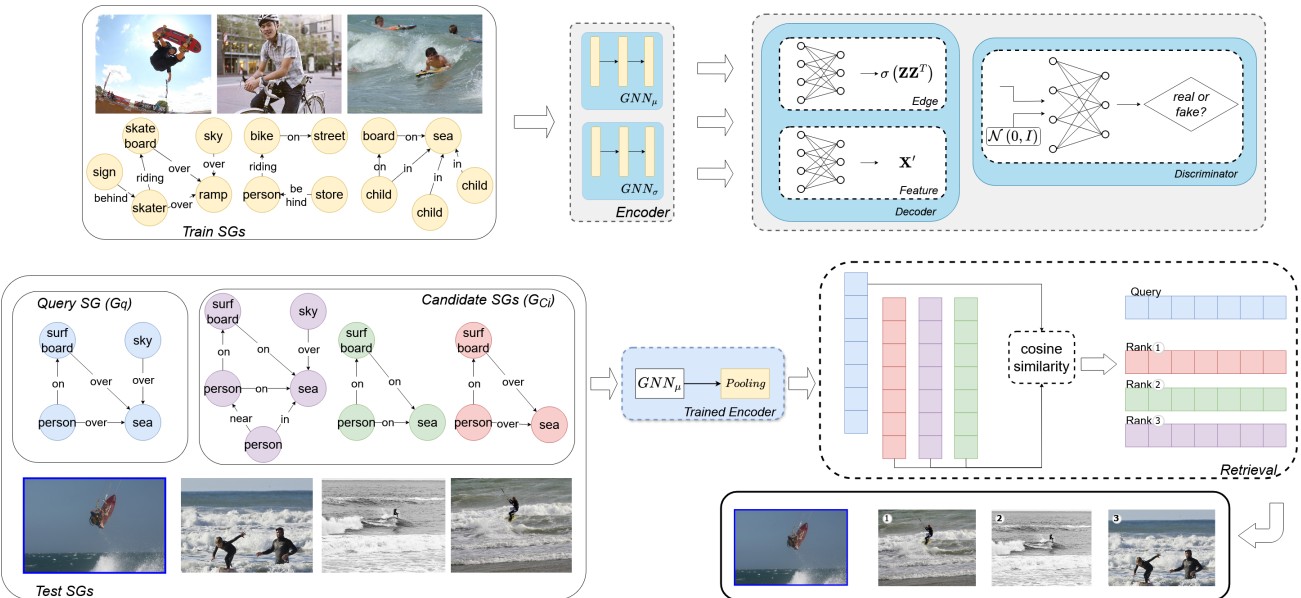

*Figure 3.* Overall Scene Graph Retrieval pipeline: training (top) and inference (bottom), with scene graphs linked to images in the dataset. The architecture of the proposed *SCENIR* model is depicted. The only loss term that does not originate from the Discriminator or the Decoder's modules is $\mathcal{L}_{KL}$ for the variational regularization, that is applied directly to the encoder output.

and outputs a latent node embeddings matrix $\mathbf{Z} \in \mathbb{R}^{n \times d_l}$. The authors of the original VGAE (Kipf & Welling, 2016) suggest leveraging a shared $GNN$ layer followed by separate $GNN_\mu$ and $GNN_\sigma$ for variational training, which is a widely adopted model option. We instead find that it is more beneficial to completely split $GNN_\mu$ and $GNN_\sigma$ into two independent 3-layer branches.

The split between $GNN_\mu$ and $GNN_\sigma$ branches is motivated by the distinct roles of mean and variance parameters in variational inference, where mean embeddings capture structural features while variance embeddings model uncertainty. Thus, the independent branches allow these distinct aspects to be learned more effectively. Ablation studies (offered in App. D) showcase that this architectural choice enhances retrieval accuracy compared to the shared-layer approach. Formally, the encoder is expressed as:

$$\begin{aligned} \mathbf{Z}_\mu &= GNN_{\mu,3}\left(GNN_{\mu,2}\left(GNN_{\mu,1}\left(\mathbf{X}, \mathbf{A}\right)\right)\right) \\ \mathbf{Z}_\sigma &= GNN_{\sigma,3}\left(GNN_{\sigma,2}\left(GNN_{\sigma,1}\left(\mathbf{X}, \mathbf{A}\right)\right)\right) \end{aligned} \quad (1)$$

All GNN functions can be implemented using any message-passing module, such as GCN (Kipf & Welling, 2017) or GIN (Xu et al., 2019). Bias terms and RELU activation functions are included for each GNN layer in the implementation. Notably, $\mathbf{Z}_\sigma$ is only employed during training.

**Decoder** A key innovation in our architecture is the decoder design, which comprises two parallel branches: an Edge Decoder and a Feature Decoder, both implemented as 2-layer MLPs instead of the traditional no-parameter inner-

product decoder, as in VGAE and ARVGA (Pan et al., 2018). This design choice allows the autoencoder to learn more sophisticated relations in the encoded latent space compared to a simple inner-product, which proves to have great impact in the case of semantic similarity between scene graphs. Mathematically, the formulation of the decoder is:

$$\begin{aligned} \mathbf{Z}_e &= W_{e,2} W_{e,1} \mathbf{Z} \\ \mathbf{A}_p &= \sigma(\mathbf{Z}_e \mathbf{Z}_e^T) \\ \mathbf{Z}_f &= W_{f,2} W_{f,1} \mathbf{Z} \end{aligned} \quad (2)$$

where $W_*$ are learnable weights (bias and RELU activation included in the implementation), $\sigma$ is the sigmoid function, $\mathbf{A}_p \in \mathbb{R}^{n \times n}$, with $\mathbf{A}_{p,ij} \in [0, 1]$, is the predicted adjacency matrix and $\mathbf{Z}_f \in \mathbb{R}^{n \times d}$ is the predicted feature matrix.

The incorporation of MLP instead of GNN in the decoder addresses the challenge of embedding oversmoothing that occurs when stacking multiple graph convolution layers (Li et al., 2018). Our choice of a 2-layer MLP decoder, and 3-layer GNN encoder architecture additionally aligns with recent findings (Luo et al., 2024) that demonstrate the effectiveness of simpler, shallow architectures over complex ones in the majority of graph-based tasks. This balanced design achieves both model expressiveness and computational speed, demonstrating superior performance in graph retrieval, while also maintaining significant computational efficiency (further details in Section 4.4).

**Discriminator** To enhance learned representation quality, we integrate adversarial training through a discriminator

module, as proposed by Pan et al. (2018). This module distinguishes between real samples from a prior distribution (typically Gaussian $\sim \mathcal{N}(0, I)$) and fake latent embeddings generated by the encoder. This adversarial component helps to further regularize the latent space and improve the overall representation quality. The discriminator is implemented as a 2-layer MLP, with a binary output (real/fake).

**Training** The model is trained end-to-end with a comprehensive loss function that combines multiple objectives from all the aforementioned modules:

$$\mathcal{L} = \lambda_1(\mathcal{L}_{feat\_recon} + \mathcal{L}_{edge\_recon}) + \\ \lambda_2\mathcal{L}_{adv} + \lambda_3\mathcal{L}_{KL} \tag{3}$$

where $\mathcal{L}_{feat\_recon}$ is the Mean Squared Error (MSE) loss between the original $\mathbf{X}$ and predicted $\mathbf{Z}_f$ feature matrices, while $\mathcal{L}_{edge\_recon}$ is the reconstruction loss between the original $\mathbf{A}$ and the predicted $\mathbf{A}_p$ adjacency matrices defined as $\mathbb{E}_{q(\mathbf{X}|\mathbf{Z}_e, \mathbf{A})}[log\, p\,(\mathbf{A}|\mathbf{Z}_e)]$. $\mathcal{L}_{adv}$ is the adversarial loss implemented as Binary Cross Entropy Loss for the discriminator prediction, and $\mathcal{L}_{KL}$ is the Kullback-Leibler Divergence of the latent embeddings to the prior Gaussian distribution $\mathcal{N}(0, I)$. To further improve training stability, we implement an Exponential Learning Rate Scheduler with a gamma of 0.95, and we apply loss tradeoff terms $\lambda_i$ (details in Appendix D) resulting in the final loss equation 3.

**Inference** At inference time, global scene graph representations are generated by applying sum-pooling to the latent node embeddings $\mathbf{Z}_\mu$ from the trained $GNN_\mu$ encoder (Fig. 3). This pooling strategy is selected for its proven superior discriminative power (Xu et al., 2019) in capturing graph-level representations (especially in cases with distinct node types, such as in scene graph representations), when compared to other pooling functions (e.g. mean or max).

## 4. Experiments

**Datasets** For our experiments we leverage the **PSG** scene graph dataset (Yang et al., 2022), a more curated version of the traditional scene graph dataset, Visual Genome (Krishna et al., 2017), as it is based on more advanced panoptic segmentation masks, containing almost $49K$ annotated image, caption and scene graph samples. We select $11K$ scene graphs for training, and $1K$ scene graphs for testing. Statistics for the finally preprocessed scene graphs are presented in Fig. 4. We also experiment on images from **Flickr30K** (Young et al., 2014) to evaluate SCENIR in a real-world use case, where caption and scene-graph annotations are unavailable, requiring us to generate synthetic ones (details about dataset preprocessing in Appendix A).

**Ground Truth and Evaluation** We employ approximate GED as the ground truth distance/similarity for evaluating

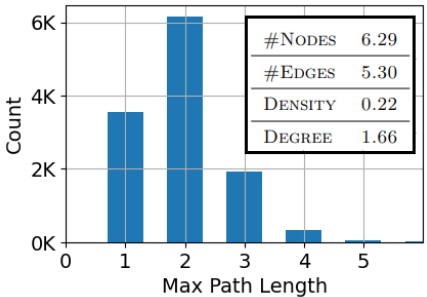

*Figure 4.* Maximum path lengths and mean values for graph metrics, for the preprocessed PSG graphs.

our approach, motivated by recent work adjacent to our field (Dimitriou et al., 2024) that emphasizes semantic similarity over low-level features, such as pixels. In accordance to their analysis and our experimental findings, GED's robustness eliminates ambiguity in generating golden rankings, unlike methods such as captioning (Fig. 2). Our experiments are based on an *inductive retrieval* setting, i.e. both the test query and candidates are **not** available to the model during training. We compute GED between all pairs of the $1K$ test graphs, and finally extract the ground-truth rankings. In total, we have $1K$ test queries, each accompanied by 999 retrieved and ranked objects. Using these GED rankings as ground truth, we evaluate all models on NDCG@k, MAP@k, and MRR ($k = 1, 3, 5, 10$).

**Baselines** We initially compare our proposed architecture, to SotA pre-trained Vision and Vision-Language (VL) models, supervised GNNs, and basic GAEs. For the Vision models, we use two pre-trained convolution-based architectures, ConvNeXt-V2-Large (Woo et al., 2023) and InceptionNeXt-Base (Yu et al., 2024), two pretrained ViT-based models, EfficientViT-L3 (Cai et al., 2023) and DeiT-III-Large (Touvron et al., 2022), and four pre-trained VL models, CLIP (ViT-L-14) (Radford et al., 2021), BLIP (base) (Li et al., 2022), BLIP-2 (COCO finetuned) (Li et al., 2023b) and AL-BEF (base) (Li et al., 2021). For the inference stage of vision models, we extract the pooled last-layer feature vectors per image, while for the VL models we encode the query image and the candidate captions for image-text retrieval. The final rankings are calculated via cosine similarity between these vectors.

Regarding GNNs, we compare with IRSGS-GCN/GIN - the current SotA for supervised scene graph retrieval - retaining the original training details (Yoon et al., 2021). We also maintain caption similarity labels as a supervision signal for training, as originally proposed by the authors, harnessing two SBERT models (MPNet and RoBERTa), in order to assess whether SBERT disagreements affects the models' output. As for the **GAEs**, we evaluate VGAE and ARVGA,

*Table 1.* Retrieval results (as occurring with comparison to ground-truth GED ranks) using supervised and unsupervised GNNs, as well as Vision & VL models. **Bold** denotes the best overall result, underlined denotes best results within each model category.

| | MODEL | NDCG↑ | | | | MAP↑ | | | | MRR↑ |
|---|---|---|---|---|---|---|---|---|---|---|
| | | @1 | @3 | @5 | @10 | @1 | @3 | @5 | @10 | |
| VISION | CONVNEXT | 12.33 | 12.50 | 12.47 | 13.06 | 24.60 | 34.87 | 36.83 | 35.33 | 41.74 |
| | INCEPTIONNEXT | 12.63 | 12.67 | 12.90 | 13.64 | 23.90 | 33.92 | 36.03 | 35.33 | 41.27 |
| | EFFICIENT-VIT | 13.47 | 13.49 | 13.40 | 13.89 | 25.70 | 35.86 | 37.27 | 36.36 | 42.66 |
| | DEIT-III | 12.75 | 13.25 | 13.05 | 13.78 | 25.40 | 35.74 | 37.32 | 36.27 | 42.48 |
| VL | CLIP | 15.63 | 14.64 | 14.58 | 14.93 | 28.80 | 38.21 | 39.38 | 38.25 | 44.79 |
| | BLIP2 | 13.64 | 14.23 | 14.60 | 15.29 | 25.90 | 36.62 | 39.14 | 38.13 | 43.73 |
| | BLIP | 15.62 | 15.19 | 15.00 | 15.38 | 28.50 | 38.57 | 40.01 | 38.86 | 45.10 |
| | ALBEF | 15.77 | 15.37 | 15.36 | 15.69 | 28.10 | 38.58 | 39.99 | 38.73 | 45.11 |
| EXISTING GNN | IRSGS-GCN$_{MPNet}$ | 27.50 | 25.84 | 24.70 | 23.53 | 41.50 | 51.33 | 51.82 | 49.18 | 56.42 |
| | IRSGS-GIN$_{MPNet}$ | 29.83 | 27.68 | 26.75 | 26.04 | 44.10 | 53.17 | 53.64 | 50.49 | 58.73 |
| | IRSGS-GCN$_{Roberta}$ | 28.93 | 26.03 | 25.20 | 24.17 | 44.10 | 53.01 | 53.62 | 50.05 | 58.19 |
| | IRSGS-GIN$_{Roberta}$ | 29.64 | 27.96 | 27.13 | 26.00 | 43.80 | 54.12 | 54.22 | 50.67 | **59.16** |
| | VGAE-GCN | 27.30 | 25.68 | 24.77 | 23.91 | 40.90 | 50.63 | 51.11 | 48.35 | 55.76 |
| | VGAE-GIN | 27.27 | 26.26 | 25.27 | 24.41 | 41.80 | 50.84 | 51.43 | 48.73 | 56.58 |
| | ARVGA-GCN | 26.59 | 25.10 | 24.26 | 23.84 | 39.70 | 49.46 | 50.07 | 47.44 | 55.09 |
| | ARVGA-GIN | 25.09 | 24.93 | 24.36 | 24.08 | 40.20 | 50.53 | 51.34 | 48.71 | 55.92 |
| OURS | SCENIR-GCN | 26.42 | 25.05 | 23.92 | 22.55 | 39.30 | 48.37 | 48.80 | 46.61 | 53.61 |
| | SCENIR-GIN | **31.39** | **28.77** | **27.59** | **26.28** | **44.60** | **54.16** | **54.27** | **51.70** | 59.01 |

as well as our SCENIR, implementing each one with GCN and GIN modules to ensure a fair comparison.

**Implementation** We utilize PyTorch Geometric (Fey & Lenssen, 2019) for supervised and unsupervised GNNs, training them on a single P100 GPU. The mean-pooled node embeddings of the last layer serve as the graph embeddings for IRSGS variants, while sum-pooling on the latent node embeddings (encoder output) is used for the GAEs. For Vision/VL pre-trained model implementations, we leverage open-source libraries (Wightman, 2019; Li et al., 2023a). For all models, predicted rankings are obtained via cosine similarity through one-step retrieval (no secondary pre-ranking), to isolate the model's retrieval abilities. More details are presented in Appendix C.

### 4.1. Quantitative Results

In Table 1 we present test set retrieval results, comparing Vision and VL models, supervised and unsupervised GNNs.

First and foremost, there is a clear performance gap between Vision models and GNNs, with Vision models scoring about half as much in NDCG and 15%-20% less in MAP and MRR. Among Vision models, ViT-based architectures (Efficient-ViT, DeiT-III) consistently outperform convolution-based ones. While VL models show modest improvements of 1%-3% over pure Vision ones, both categories fall significantly behind GNN-based architectures on scene graph (and equivalently image-to-image) retrieval.

Focusing on the GNNs, we observe that the supervised IRSGS performs competitively over the basic unsupervised VGAE and ARVGA, holding a consistent advantage across all metrics. Notably, the GIN variants consistently outperform their GCN counterparts for all architectures, which can be attributed to the theoretically proven expressive power of the GIN module (Xu et al., 2019). Indeed, our SCENIR-GIN variant achieves 2-3% increase in NDCG, 3-5% in MAP, and 2-5% in MRR compared to its GCN variant. This pattern holds for both supervised and unsupervised approaches, suggesting that GIN's structural advantages generalize across different training paradigms.

Overall, our proposed SCENIR architecture demonstrates superior performance, surpassing even the supervised IRSGS across most metrics. Our comprehensive unsupervised approach, combining an MLP-based decoder with a powerful GIN backbone, can effectively substitute for the absence of training labels while providing SotA retrieval.

**Ablation Studies** To understand the contribution of each integrated architectural component, we conduct ablation experiments (Table 2), gradually subtracting the main constituents analyzed in Section 3.2. As evidenced in Table 2, the full benefit of the selected architectural components is only realized when properly integrated within our entire proposed framework. For example, excluding the MLP decoder immediately hurts performance by ∼4% in NDCG, MAP and MRR, while further removing the discriminator slightly

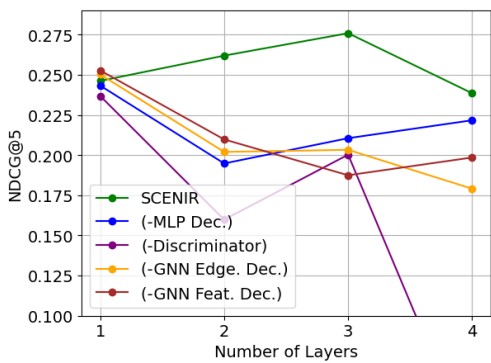

*Figure 5.* NDCG@5 score for different number of GNN layers.

deteriorates results. However, when subtracting the GNN Feature Decoder and the GNN Edge Decoder we observe some marginal gains, without however approximating the full SCENIR performance. Additional experiments with varying numbers of GNN layers in Figure 5 (further results in Appendix D) show that SCENIR uniquely benefits from a deeper architecture, achieving optimal performance with 3 layers, while other models exhibit decreased performance beyond a single layer. This aligns with the path lengths of our scene graphs (Fig. 4), allowing SCENIR to achieve semantic representation without over-smoothing.

*Table 2.* Ablation study results showing the impact of different modules for SCENIR. Higher values are better for all metrics.

| MODEL | NDCG@3↑ | MAP@3↑ | MRR↑ |
|---|---|---|---|
| **SCENIR** | **28.27** | **54.16** | **59.01** |
| (-MLP DEC.) | 24.95 | 49.27 | 54.95 |
| (-DISCRIMINATOR) | 24.37 | 49.02 | 54.22 |
| (-GNN EDGE DEC.) | 25.81 | 50.99 | 56.61 |
| (-GNN FEATURE DEC.) | 25.68 | 50.63 | 55.76 |

### 4.2. Qualitative Results

Some qualitative results that demonstrate the superiority of our approach are presented in Figures 6 and 7. In Figure 6, DeiT, a VL model, completely fails to retrieve a relevant image to the query, which depicts a black cat wearing a white bow tie. Derailed by color-related black& white patterns, DeiT returns a boy wearing a tuxedo in top-1 position. Failed retrievals follow in the top-2 and top-3 places, portraying people in athletic situations, the semantics of which are totally unrelated to the query ones. It is uncertain how DeiT decision-making works in these situations, since the query image accurately falls within DeiT's pre-training distribution on ImageNet-1K (Deng et al., 2009) (which includes cat classes, denoting that related features should be well-imbued within DeiT's representation). Regarding

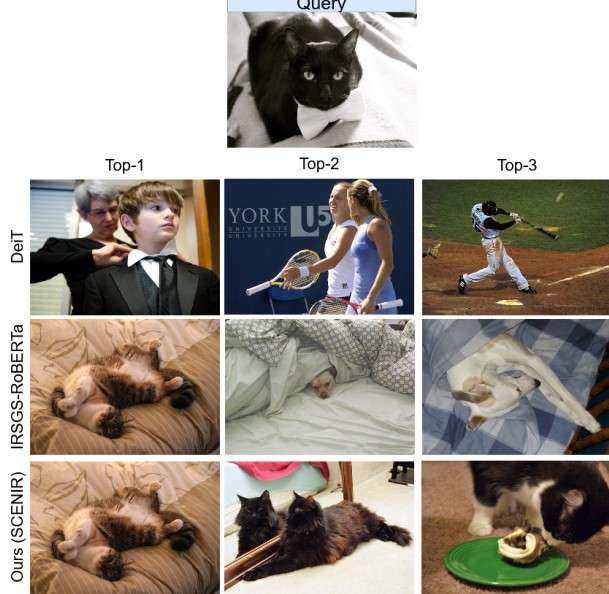

*Figure 6.* Qualitative results: VL (DeiT) **vs** supervised GNNs (IRSGS-GIN w RoBERTa-based caption similarity) **vs** SCENIR.

supervised GNNs, even though the top-1 result displaying a cat is reasonable, subsequent ones depict dogs instead of cats, revealing the inability of the best IRSGS variant (with GIN and RoBERTa for caption embeddings) to return the key semantics of the query. Still, the semantic understanding is elevated in comparison to DeiT, with displayed concepts staying within the 'animal' category. On the contrary, SCENIR successfully returns cat images in all top-three positions, also qualitatively surpassing its competitors.

**Caption-driven disagreements** In Fig. 7, the impact of disagreeing SBERT captions and in turn ground-truth ranks becomes prominent, as proven by the diverging outputs of IRSGS variants. Specifically, RoBERTa and MPNet-based supervision signals lead to a discrepancy in all positions, with the MPNet-based one also returning an irrelevant image at top-2, which does not contain any of the 'bird' or 'bench' query semantics. Other than the compared supervised GNNs, EfficientViT, lying in the Vision models category, is also entirely confused in top-2 and top-3 positions, resulting in a striking failure to bring images containing at least one relevant semantic. On the contrary, SCENIR successfully retrieves bird-related images in all three positions.

### 4.3. Extendability

**In-the-wild Retrieval** To validate our model's practical applicability, we evaluate SCENIR's performance on images from Flickr30K. This represents a more challenging real-world scenario where ground truth scene graphs, or captions are unavailable. We employ PSGTR (Yang et al., 2022)

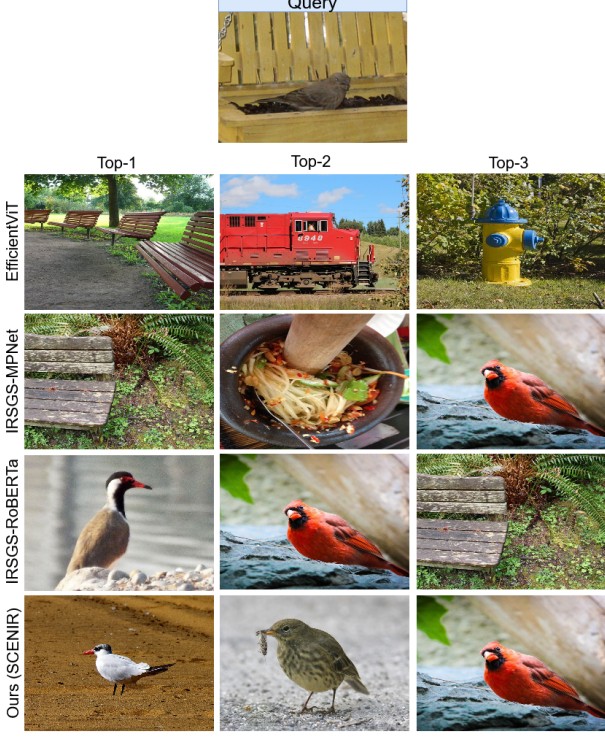

*Figure 7.* Additional qualitative results. Inconsistent ground truth matchings lead to inconsistent IRSGS-GIN outputs.

for automated scene graph generation and BLIP-Captioner-Base (Li et al., 2022) for caption generation to process the raw images. As shown in Table 3, SCENIR maintains its superior performance even in this challenging setting, achieving the *highest scores across all retrieval metrics*. These results demonstrate SCENIR's robustness and generalizability beyond curated datasets, rendering it particularly valuable for real-world scene graph retrieval applications.

*Table 3.* Image Retrieval without pre-annotated scene graphs and captions on Flickr30K.

| MODEL | NDCG@3↑ | MAP@3↑ | MRR↑ |
|---|---|---|---|
| IRSGS | 21.02 | 49.08 | 54.30 |
| VGAE | 17.92 | 44.81 | 50.45 |
| ARVGA | 17.51 | 44.17 | 49.54 |
| SCENIR | **22.75** | **50.31** | **56.20** |

**Counterfactual Retrieval** Retrieval with emphasis on depicted semantics - objects and relations - is essential in tasks such as counterfactual explanations (CEs) (Browne & Swift, 2020). Semantic-driven CE frameworks identify the minimum changes required to transit between classification labels. For image classifiers, this involves finding the most similar image from a different class and computing the ed-

*Table 4.* Counterfactual Scene Graph Retrieval on PSG.

| MODEL | NDCG(B)@1↑ | MAP(B)@3↑ | MRR(B)↑ |
|---|---|---|---|
| VGAE | 8.79 | 11.09 | 14.23 |
| ARVGA | 8.2 | 10.63 | 13.79 |
| IRSGS | 8.9 | 11.63 | 14.96 |
| GC | 7.0 | 8.7 | 10.67 |
| SCENIR | **9.7** | **11.83** | **14.99** |

its between them; particularly, graph edits between scene graphs to instruct CEs are suggested in the SotA framework of **G**raph **C**ounterfactuals (**GC**) (Dimitriou et al., 2024). Due to the critical role of the scene graphs in this use case, we demonstrate the versatility of SCENIR when adopted as a retrieval component in a CE pipeline: given a query scene graph in class $A$ we seek the most similar scene graph of class $B \neq A$ according to Places365 classifier (Zhou et al., 2017), which ultimately serves as the CE between the corresponding images. For evaluation metrics, we utilize the binary versions (denoted by 'B'), introduced in **GC**, where only the top-1 ground-truth instance is considered relevant[2]. Unlike GC's supervised transductive approach that requires test graphs during training, our evaluation follows a more challenging and realistic *inductive setting* with completely unseen test graphs. As shown in Table 4, SCENIR *outperforms all supervised and unsupervised baselines* along with the previous SotA GC framework (which is optimized for calculating CEs) across all metrics, illustrating our superiority in counterfactual retrieval, without requiring explicit supervision or exposure to the test set during training.

*Table 5.* Complexities of scene graph retrieval frameworks, with respect to the dataset size (PSG dataset, 11k/1k train/test graphs).

| MODEL | PREPROC. | TRAINING | INFERENCE | TOTAL TIME |
|---|---|---|---|---|
| GC | $\mathcal{O}(n^2)$ | $\mathcal{O}(n^2)$ | $\mathcal{O}(n)$ | ∼3 hr. |
| IRSGS | $\mathcal{O}(n^2)$ | $\mathcal{O}(n)$ | $\mathcal{O}(n)$ | ∼50 min. |
| SCENIR | $\mathcal{O}(n)$ | $\mathcal{O}(n)$ | $\mathcal{O}(n)$ | ∼**8 min.** |

### 4.4. Computational speedup

As shown in Table 5, SCENIR achieves *significant computational advantages* over competitors. Prior SotA supervised methods like IRSGS and GC require quadratic complexity with respect to the sampled PSG dataset size in preprocessing and/or training with runtimes of ∼50 minutes and ∼3 hours respectively. SCENIR though maintains **linear complexity**, executing the entire pipeline in just ∼8 minutes. This efficiency stems from our unsupervised approach that eliminates the need for expensive preprocessing of similarity

---

[2]The CE setting instructs that **only** the closest instance is relevant, due to minimality constrains of counterfactual theory.

labels or caption embeddings. The linear complexity across preprocessing, training, and inference makes SCENIR particularly suitable for large-scale retrieval applications.

## 5. Conclusion

In this work, we propose SCENIR, a novel unsupervised framework for scene graph retrieval based on graph autoencoders. We emphasize the importance of using Graph Edit Distance, a deterministic graph similarity algorithm, for evaluating scene graph retrieval. Our proposed model achieves superior performance in retrieval metrics when compared to SotA Vision and VL models, as well as supervised GNNs, while also being significantly faster than its GNN competitors. The qualitative results showcase the robustness and effectiveness of our approach in retrieving relevant images, contributing to the underexplored domain of scene graph retrieval and highlighting the potential of unsupervised approaches in this field. Through our extendability experiments on unannotated datasets and counterfactual explanation applications, we further highlight the broad applicability of our framework in real-world scenarios.

## Acknowledgments

This work was supported by the Hellenic Foundation for Research and Innovation (HFRI) under the 5th Call for HFRI PhD Fellowships (Fellowship Number 19268). We thank all reviewers for their insightful comments and feedback.

## Impact Statement

This paper presents work whose goal is to advance the field of Machine Learning and Image-to-Image Retrieval. There are no societal consequences of our work concerning the utilization of data such as image datasets, which are publicly available and of general use.

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

## A. Dataset Preprocessing

For the preprocessing of the PSG dataset, we removed the isolated nodes (no incoming and no outgoing edges) of all the scene graphs, mainly for two reasons: Firstly, the isolated nodes do not contribute in any way during the propagation through graph convolutional layers (neither do they provide, nor receive any information). Secondly, the PSG dataset defines a special type of grouped node (noted as *"merged"*), which basically represents multiple nodes of the same type. We only retain these merged nodes, implicitly keeping all the important information (object types and interactions), without obstructing the training process of the GNNs. Examples of the final scene graphs can be seen in Fig. 11. We used 768-dimensional Sentence-Transformer embeddings for the 189 object and predicate classes to construct the feature matrix $\mathbf{X}$ for each scene graph, which is required as input for all the GNN models.

## B. Ground Truth and Retrieval Metrics

For the computation of the ground-truth GED scores, we need to define costs for each edit operation on the input graphs (insertion,deletion and substitution). Specifically, we define the node substitution cost as the cosine distance between the node embeddings, while the node insertion/deletion cost is defined as the cosine distance to the mean node embedding (average of all the 133 object class embeddings). Additionally for the retrieval metrics, we focus on the top-50 items out of the 999 retrieved by GED to avoid inflated MAP and MRR scores. For NDCG, we scale the inverse GED score to a range of $[1, 10]$ to manage outlier instabilities.

It is worth noting that the retrieval metrics behave differently, when changing the number of top-$k$ retrievals. Specifically, we can see in Table 1 that NDCG steadily decreases as $k$ goes from 1 to 10. This is expected, as it is a harder task to have a perfect top-$m$ retrieval, than a perfect top-$k$ retrieval, where $m > k$. This is not the case with MAP, where we have significantly lower scores when $k = 1$, compared to $k = 3, 5, 10$. This exception is a direct result of the definition of MAP, because it only considers whether a retrieved object is relevant or not. It does not use any relevance scoring system, like NDCG, something that renders it vulnerable to situations where the top retrieved items vary in relevance. In this case, if there are highly relevant retrieved top-1 items, NDCG increases according to their relevance score, but MAP will remain constant independent of their relevance. This phenomenon is especially evident with MAP@1, because the final score is entirely determined by a single retrieved item.

## C. GNN Training Details

Regarding the *Graph Autoencoders*, they were all trained for 30 epochs, with batch size 64, AdamW Optimizer ($lr = 0.001$, $\beta_1 = 0.9$, $\beta_2 = 0.999$, $weight\_decay = 0.01$), 1000 latent space dimension, 32 output dimension for the Edge Decoder, 768 output dimension for the Feature Decoder, and 1 output dimension for the Discriminator (real/fake). Concerning the models that employ adversarial training, we followed the training algorithm proposed in Pan et al. (2018), with two separate AdamW optimizers, one for the Discriminator, and one for the rest of the model parameters. Also, we used an Exponential Learning Rate Scheduler ($\gamma = 0.95$), and Loss Tradeoff terms in order to stabilize the training. Experiments for Loss Tradeoff terms can be found in the next section.

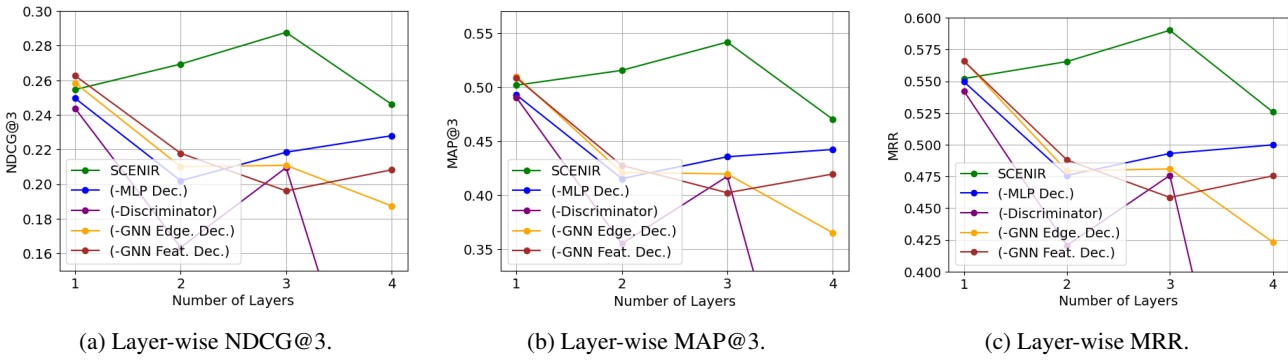

(a) Layer-wise NDCG@3.    (b) Layer-wise MAP@3.    (c) Layer-wise MRR.

*Figure 8.* Variation in NDCG@3, MAP@3 and MRR for different number of layers, for each GAE. The 3-layer variant of SCENIR performs the best in every metric.

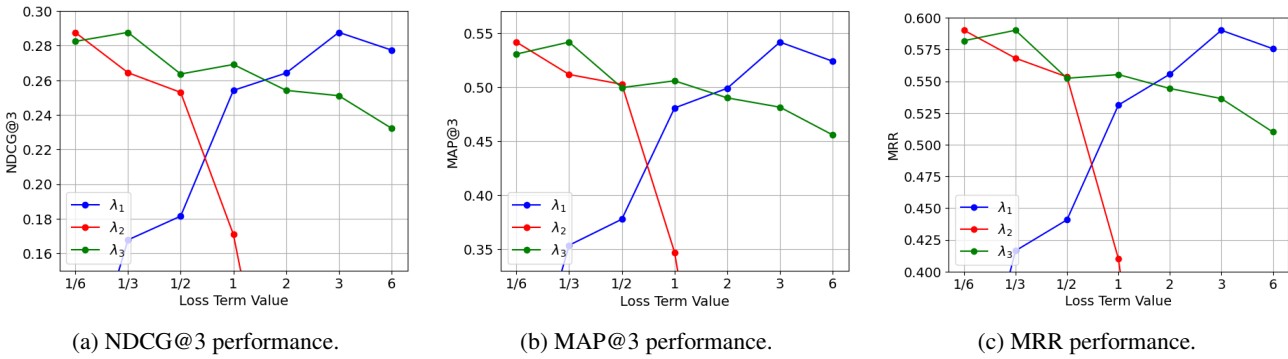

| (a) NDCG@3 performance. | (b) MAP@3 performance. | (c) MRR performance. |

*Figure 9.* Variation in NDCG@3, MAP@3 and MRR for different values of each loss term, in the final loss function.

## D. Further GNN Architecture Experiments

In Figure 8, we report the results on NDCG@3, MAP@3 and MRR for different number of **GNN layers**, on the SCENIR-GIN architecture. Results follow the same trajectory depicted in Fig. 7 of the main paper, showing that the 3-layer GNN outperforms the rest, on every metric.

We also report the results on **Loss Trade-off Term** tuning. Specifically, the final loss function of SCENIR, can be written as:

$$\mathcal{L} = \lambda_1 \left( \mathcal{L}_{feat\_recon} + \mathcal{L}_{edge\_recon} \right) + \lambda_2 \mathcal{L}_{adv} + \lambda_3 \mathcal{L}_{KL} \tag{4}$$

Here, $\lambda_1$ denotes the loss weight for the graph reconstruction, $\lambda_2$ the loss weight for adversarial regularization, and $\lambda_3$ the loss weight for KL Divergence. Each loss term was individually tuned by fixing the other two to their empirically determined optimal values and varying the remaining term to identify its best value. The final chosen parameters where $\lambda_1 = 3$, $\lambda_2 = \frac{1}{6}$ and $\lambda_3 = \frac{1}{3}$. Results for NDCG@3, MAP@3 and MRR of the SCENIR-GIN model, reported in Figure 9.

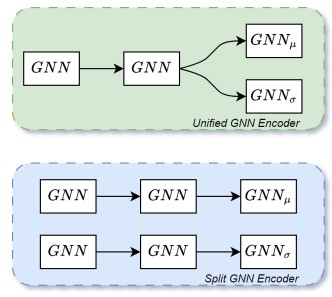

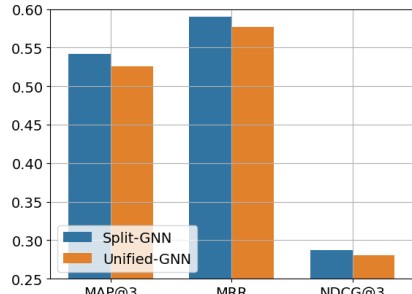

*Figure 10.* Left: illustration of the encoder architecture in original GAE (unified), and in the proposed SCENIR (split). Right: Performance comparison between split and unified architecture, on MAP@3, MRR and NDCG@3.

Finally, regarding the splitting of the GNN Encoder, the original GAE (Kipf & Welling, 2016), proposed a unified encoder architecture (as seen in Figure 10), where all the layers are shared, except for the output. We proposed using a completely Split variation for the GNN encoder, without any shared layers, which ultimately surpasses the original variant across every metric. Architecture illustration and metrics results can be seen in Figure 10.

## E. Preprocessed Scene Graphs of Qualitative Results

In Figures 12 and 13, we present the scene graphs accompanying the qualitative results of Figures 5 and 6 of the main paper respectively. Viewing the graphs with the image retrieval outcomes provides a better understanding of our graph-based semantic retrieval system and its merits. In both cases, it is apparent that GNN-based models (supervised and unsupervised) retrieve structurally similar graphs varying in semantic content; our method leads to consistently better semantics.

It is important to highlight that PSG graphs have been curated so that they properly represent semantics and relationships

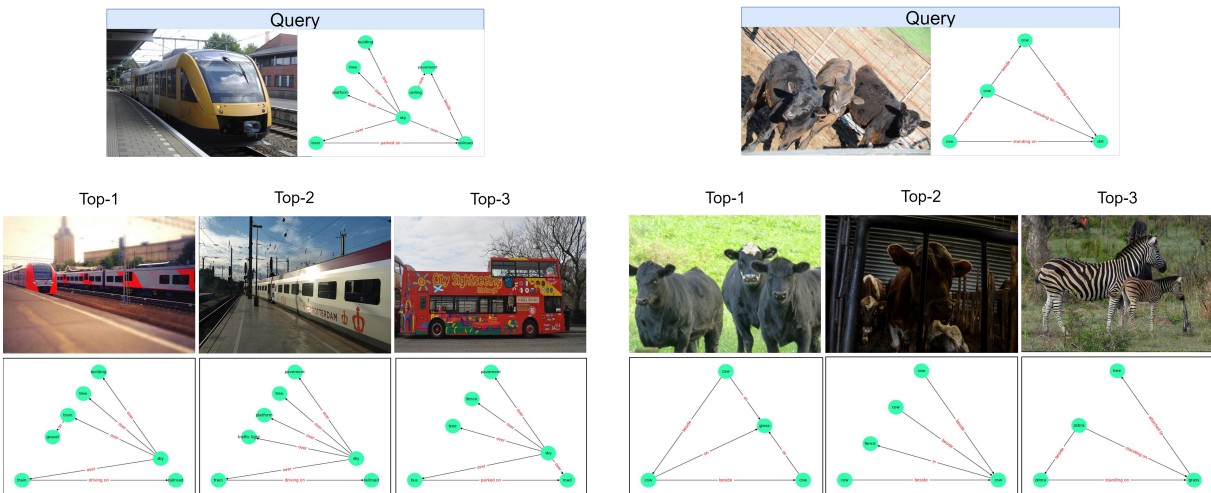

*Figure 11.* Additional qualitative results obtained from SCENIR. Scene graphs of retrieved images are also provided.

present in images. To this end, our graph based method does not favor scene graphs being similar to the query because of merely having similar annotations; on the contrary, it is ensured that similarity in terms of scene graph annotations denotes actual similarity in terms of semantics and relationships.

## F. Additional Results

In Fig. 11 we provide some additional ranked images as retrieved from our proposed model, SCENIR.

We can observe a general semantic consistency among the retrieved results. Starting from the example on the left side of Fig. 11, we can see that our framework successfully returns semantically similar images in all three positions. Specifically, in the first two positions, we can see a train on the railroad, something that is also present in the scene graphs. For the image in the third position, while the vehicle itself is different ("bus" instead of "train"), there is still significant similarity in the corresponding scene graphs.

As for the example on the right side of Fig. 11, we can see more clearly the structural similarities of the query and the retrieved scene graphs. In all cases, semantics of nodes and edges are very similar. Specifically, in the first two positions, the retrieved images depict multiple cows, something that is also evident in the scene graphs with the "cow" nodes. Since the test set was randomly chosen, it is evident that finding exact-match images is an uncommon scenario. Therefore, the image in the third position, depicts zebras instead of cows. It is important to note that many of the semantics remain similar (both structurally and conceptually), while the concepts of "cow" and "zebra" are particularly close.

## G. Details on Semantic Counterfactual Retrieval

Counterfactual explanations provide insights into why a machine learning model made a particular decision by identifying the smallest possible changes to an input that would have led to a different outcome. For example, in the context of loan approval, a counterfactual explanation might state: "Had your income been $5,000 higher, your loan would have been approved." This method is particularly useful for interpretability, as it helps users understand not just the current decision, but what would need to change to alter it.

Semantic counterfactual explanations extend this idea by ensuring that the generated modifications are meaningful and realistic within the context of the data. While counterfactuals in general focus on any minimal change that flips the model's prediction, semantic counterfactuals ensure that these changes preserve the logic and constraints of the domain. For instance, in an image classification model, a semantic counterfactual would modify features in a way that aligns with real-world variations (e.g., changing a dog's breed rather than distorting the image with unrealistic pixel noise). Within the context of semantic graph counterfactuals proposed by Dimitriou et al. (2024), images are represented as scene graphs, and the goal is to identify all meaningful edits required to transform a scene graph belonging to one source class to the most similar scene

graph/image classified differently. Therefore, their use case can directly leverage our scene graph retrieval framework.

Their method relies on Graph Edit Distance (GED) as a supervision signal, which measures the smallest number of edit operations (node/edge additions, deletions, or label changes) required to transform one graph into another. By leveraging GED, one can generate counterfactuals by using GNNs to learn to approximate GED and thus minimize the number of modifications needed to alter the model's decision. However, computing GED is computationally expensive, particularly in a supervised learning setting where large datasets require repeated counterfactual evaluations. Even within the GC paper it is highlighted that this brute-force approach is somewhat inefficient, making it impractical for real-world applications that require fast and scalable explanations.

Our proposed framework, SCENIR, directly addresses these shortcomings. Its efficiency stems from its fully unsupervised design, which not only reduces training time but also removes dependencies on labeled data, making it more scalable for large-scale retrieval applications. This aspect is evident in the significantly smaller training and inference times, as shown in Table 5. Additionally, our approach differs in that GED is used as a ground truth measure while maintaining an unsupervised learning paradigm, allowing for a more flexible and inductive generalization.

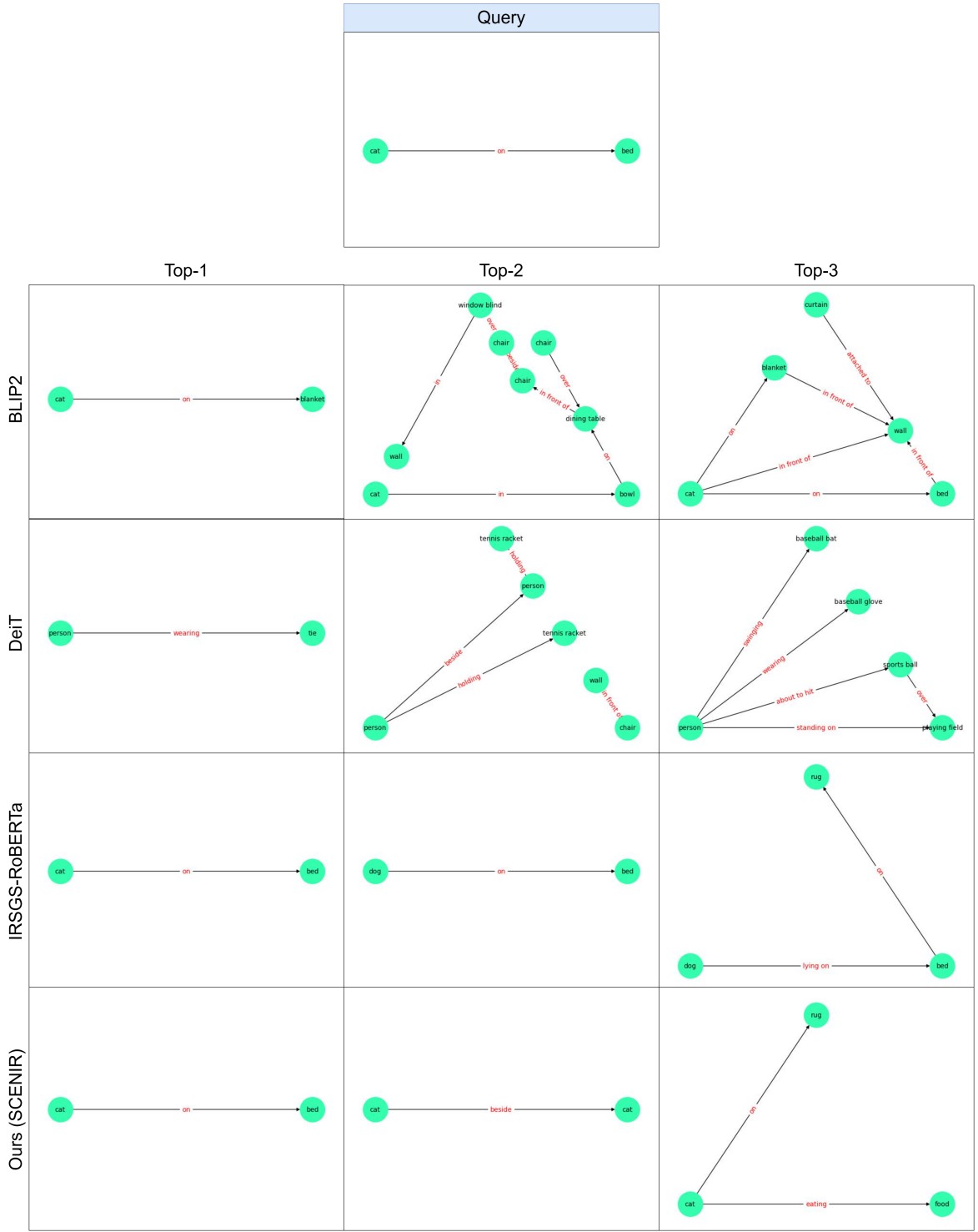

*Figure 12.* Underlying preprocessed scene graphs directly corresponding to the qualitative results of Figure 5 in the main paper.

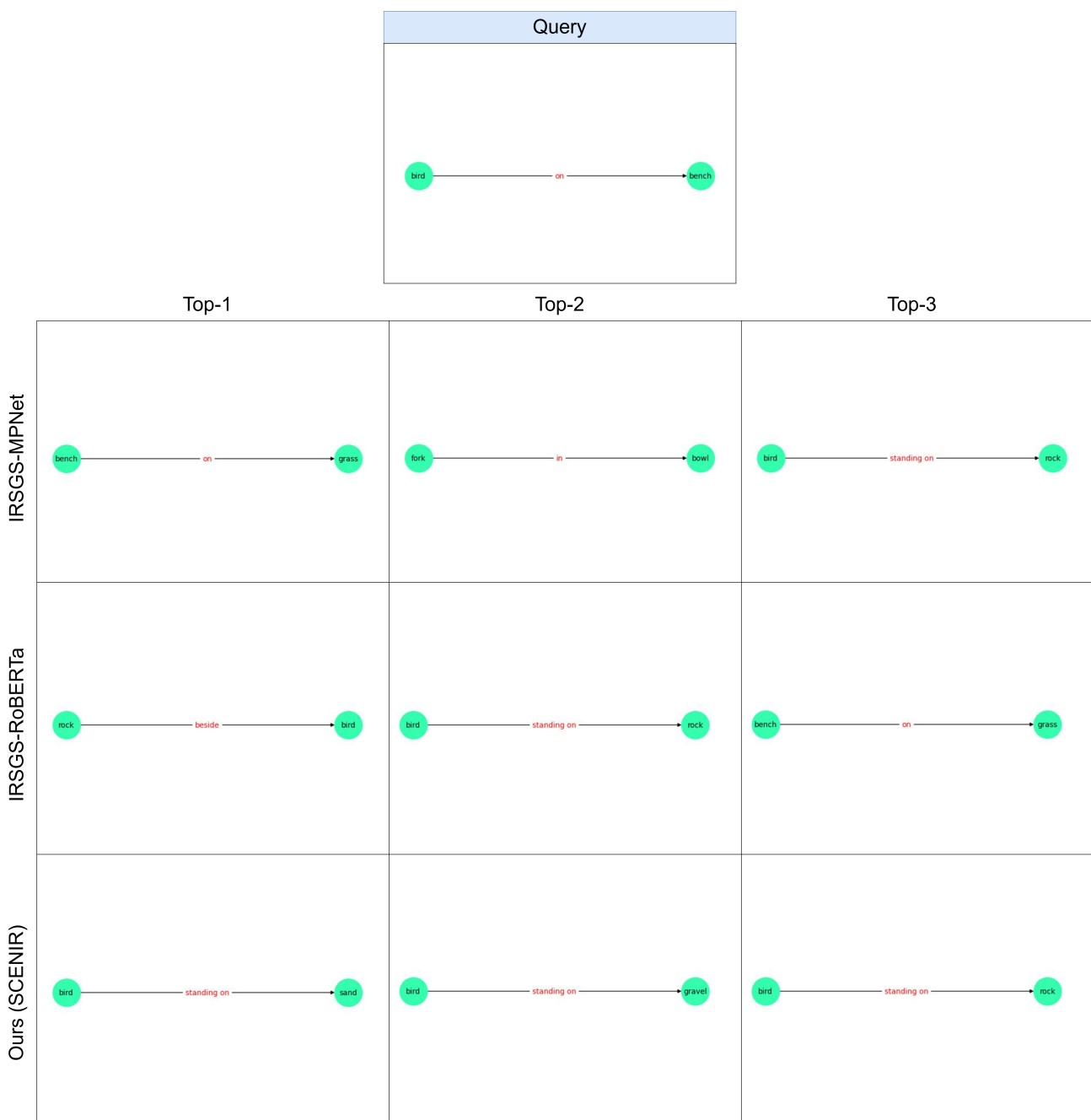

*Figure 13.* Underlying preprocessed scene graphs directly corresponding to the qualitative results of Figure 6 in the main paper.

