# OpenReview forum: "SCENIR: Visual Semantic Clarity through Unsupervised Scene Graph Retrieval"
_ICML.cc/2025/Conference — ICML 2025 poster_

### Official Review · Reviewer_PwzX · 2025-03-10

**Overall Recommendation:** 3

**Summary:**

SCENIR is a novel unsupervised scene graph-based retrieval framework that prioritizes semantic content over low-level visual features. SCENIR uses a Graph Autoencoder to eliminate the need for labeled data. It outperforms vision-based, multimodal, and supervised GNN approaches in both accuracy and efficiency. Additionally, it introduces Graph Edit Distance (GED) as a robust metric for scene graph similarity, improving retrieval reliability and enabling generalization to unannotated datasets for counterfactual image retrieval.

**Claims And Evidence:**

Yes

**Essential References Not Discussed:**

It seems that essential references are discussed.

**Experimental Designs Or Analyses:**

Yes

**Methods And Evaluation Criteria:**

Yes

**Other Comments Or Suggestions:**

Refer to the weaknesses mentioned above

**Other Strengths And Weaknesses:**

S1: SCENIR overcomes visual biases in vision models by using scene graphs to enhance semantic understanding.

S2: It introduces an unsupervised Graph Autoencoder for scene graph retrieval, removing the need for labeled data.

S3: SCENIR ensures robust evaluation with GED and extends to unannotated datasets for broader applications.

W1: Figure 1 could use clearer examples, as the Top-1 result of Efficient-ViT appears more similar to SCENIR's Top-1 result, where the sports-related semantic relationship is not prominent.

W2: Clarify the rationale for using GED as an evaluation metric in scene graph retrieval, emphasizing its ability to capture structural and semantic differences effectively.

W3: The paper could further explore or compare the capabilities of recent large language and multimodal models in addressing the scene graph retrieval problem.

**Questions For Authors:**

Refer to the weaknesses mentioned above

**Relation To Broader Scientific Literature:**

Yes

**Theoretical Claims:**

Yes

---

> ### Author Rebuttal · Authors · 2025-03-30
>
> We sincerely appreciate Reviewer PwzX's thoughtful feedback and their recognition of the validity of our method and the soundness of our evaluation. We now provide clarifications for each of the concerns raised.
>
> - W1: We appreciate the reviewer’s suggestion regarding Figure 1 and would like to clarify that our design choices for the teaser figure were intentional. The selected images specifically illustrate the color bias present in Efficient-ViT, which SCENIR effectively mitigates. This example highlights SCENIR’s ability to retrieve images based on true semantic understanding rather than superficial color similarities. Examining the ranked results in detail (all three positions), we can observe Efficient-ViT’s color bias when analyzing the semantic relevance of the retrieved images, as discussed in lines 47-56 of the Introduction:
>     1. The **Top-1 result** from Efficient-ViT does contain a ‘group of people,’ but they are not engaged in ‘riding’ any form of ‘sports equipment’ or ‘wheeled vehicle’ on the ‘street’ - key semantic elements of the query.
>     2. While the **Top-2 result** might appear somewhat relevant due to the presence of a car, it lacks critical query concepts such as ‘group of people,’ ‘riding,’ and ‘sports equipment,’ demonstrating a reliance on color rather than semantics (as noted in lines 49-53 of the Introduction).
>     3. The **Top-3 result** from Efficient-ViT moves away from a black-and-white image but still fails to capture the essential semantics, as it lacks both a ‘group of people’ and any form of ‘sports equipment.’ In contrast, SCENIR retrieves an image of people snowboarding, which, while differing in activity, aligns more closely with the core query semantics than Efficient-ViT’s selections.
> - W2: We appreciate the reviewer’s request for clarification and would like to reiterate the rationale behind using GED as an evaluation metric, as detailed in the Ground Truth and Evaluation section (starting at line 267). GED provides a deterministic method for identifying the most similar graph pairings in a dataset by measuring structural differences. Specifically, it measures structural differences: It quantifies the dissimilarity between two graphs by counting the minimum number of edit operations (node/edge insertion, deletion, substitution) needed to transform one graph into another. This ensures that retrieval is based on meaningful changes in object compositions and relationships, rather than just superficial features. Unlike purely visual metrics (e.g., pixel similarity) or text-based evaluations (e.g., caption similarity), GED directly measures the semantic coherence between two scene graphs. Semantically related objects or objects with similar roles (e.g., "person sitting on a chair" vs. "child sitting on a bench") will have lower edit distances, reflecting semantic similarity. Motivated by these advantages, as well as non-negligible evaluation shortcomings stemming from leveraging captions and caption similarity (see inconsistent top-1 SBERT retrievals in Figure 1, as well as low agreement between varying SBERT models in Figure 2), together with the support of prior literature in scene graph similarity (Dimitriou et al, 2024), we conclude that GED arises as an ideal evaluation measure in scene graph retrieval, promoting semantic preservation while eliminating ambiguity in defining ground-truth pairings.
> - W3: We appreciate this interesting suggestion; however, we strongly believe it aligns with a parallel research direction rather than the core focus of our work. One of our main claims is computational efficiency (see Section 4.4. Computational speedup): SCENIR achieves fast retrieval by executing all stages in the pipeline in around 8 minutes, while requiring minimal computational resources (single NVidia Tesla P100 16Gb VRAM GPU). On the other hand, utilizing LLMs/Multimodal LLMs already increases the computational budget, even if we exclude the pre-training stage, since most potent LLMs require larger GPUs than the one used in our experiments, or paid APIs (e.g. as in the case of ChatGPT). Other than that, harnessing scene graphs directly fuses semantic information within the pipeline, ensuring determinism in results, meaning that multiple runs of the pipeline yield consistent results. On the other hand, multimodal LLMs introduce variability, as they are not explicitly guided on which features to prioritize, potentially leading to biases similar to those observed in visual and vision-language models (e.g., Efficient-ViT in Figure 1 and DEiT in Figure 6), along with additional prompt-based variability. In any case, we regard this comment as a future work direction.
>
> We once again thank the reviewer for their valuable feedback and hope this clarification addresses their concerns.

---

> > ### Comment · Reviewer_PwzX · 2025-04-06
> >
> > Thank you for your response and for addressing my previous comments. After careful review and consideration, I would like to keep my original score.

---

### Official Review · Reviewer_tG8k · 2025-03-13

**Overall Recommendation:** 3

**Summary:**

The paper introduces an unsupervised framework for scene graph retrieval using GNN to prioritize semantic content over low-level visual biases. It employs a graph autoencoder to learn scene graph embeddings without labeled data and advocates for Graph Edit Distance as a deterministic evaluation metric.

**Claims And Evidence:**

NA

**Essential References Not Discussed:**

NA

**Experimental Designs Or Analyses:**

NA

**Methods And Evaluation Criteria:**

method.

**Other Comments Or Suggestions:**

c.f., Weakness.

**Other Strengths And Weaknesses:**

S:
1\ Scene graphs explicitly model objects and relationships, mitigating biases from superficial features like color
2\ Extends to unannotated datasets using automated scene graph generation
3\ Introduces GED as a deterministic ground truth, addressing variability in caption-based evaluation.

W:
1\ Experiments focus on PSG and Flickr30K, broader validation across diverse domains is needed.
2\ The impact of adversarial training and decoder design could benefit from deeper analysis. This part need more ablations.

**Questions For Authors:**

What is the sensitivity of SCENIR to different scene graph generation models?

**Relation To Broader Scientific Literature:**

NA

**Theoretical Claims:**

NA

---

> ### Author Rebuttal · Authors · 2025-03-30
>
> We sincerely thank Reviewer tG8k for their thoughtful comments and for acknowledging the strengths of our approach, methodology, and evaluation process. We now address your concerns systematically
>
> - W1: To address the reviewer’s valuable concern regarding the comprehensiveness of the datasets used in our work, we offer the following clarification. Since our approach focuses on scene graphs, and after a thorough investigation of potentially related datasets, we found that PSG and Flickr30K serve as a superset or an improved version of other similar datasets (e.g., Visual Genome, as mentioned in lines 254-255). Experimenting with additional datasets like Visual Genome would lead to redundancy, as it contains the same images from MSCOCO. The richness of scenes, objects, and relationships in the PSG and Flickr30K datasets already covers a wide range of domains and scenarios, thereby eliminating the need to explore other datasets that would essentially reflect the same data distribution. Furthermore, our experiments on Flickr30K with synthetic scene graphs and PSG (or Visual Genome) with annotated scene graphs follow the experimental setup used in prior work (Yoon et al., IRSGS).
> - W2: Regarding the impact of adversarial training and the feature/edge decoder design, we would like to clarify that we have already conducted ablation studies on both components in Section 4.1. The effect of adding or removing the adversarial loss is shown in line 3 of Table 4, while the decoder ablations are detailed in lines 4 and 5 of the same table, further supporting our design choices for the final framework architecture. While we appreciate the reviewer’s perspective, given our focus on proposing a novel end-to-end semantic image retrieval framework, rather than optimizing specific GNN components, a more fine-grained ablation analysis (e.g., tuning the normalization of the adversarial training module) falls beyond the intended scope of our work.
> - Questions For Authors: Despite the irrefutable validity of such a query, we believe that evaluating SGG frameworks’ quality falls outside the scope of this work. In our study, we report findings using the most effective SGG module available, as older frameworks may struggle to accurately represent scenes, potentially limiting retrieval quality. Therefore, we recommend adapting SCENIR to the strongest SGG framework for meaningful results. That said, SGG is not a core component of the SCENIR pipeline but is included primarily to demonstrate the effortless extendability of our approach to unannotated datasets.
>
> We once again thank the reviewer for their valuable feedback and hope this clarification addresses their concerns.

---

### Official Review · Reviewer_kfau · 2025-03-14

**Overall Recommendation:** 4

**Summary:**

This paper presents SCENIR, an unsupervised framework for scene graph retrieval that aims to improve semantic understanding in image-to-image retrieval tasks. It introduces a Graph Autoencoder-based architecture, eliminating the dependence on supervised ground truth labels like captions, which suffer from variability and inconsistencies. Key contributions include advocating Graph Edit Distance (GED) as a deterministic ground-truth measure, superior retrieval performance compared to both vision and supervised GNN baselines, and demonstrated applicability to real-world datasets and counterfactual retrieval scenarios. Experimental results on PSG and Flickr show SCENIR outperforming state-of-the-art methods in terms of accuracy and computational efficiency.

**Claims And Evidence:**

The major claims, improved retrieval performance and computational efficiency, are supported by experimental results, comparisons, and ablation studies.

**Essential References Not Discussed:**

n/a

**Experimental Designs Or Analyses:**

The experimental designs are thorough and valid. Various models including Vision, Vision-Language (VL), supervised GNNs, and unsupervised GAEs were fairly compared using clear and appropriate evaluation metrics.
A slight limitation might be the lack of an ablation study on diverse graph embedding dimensions.

**Methods And Evaluation Criteria:**

The methods (GAE-based unsupervised learning with adversarial training, graph pooling strategies, GED evaluation) are suitable for addressing biases arising from low-level visual features in scene graph retrieval tasks. Using GED as the evaluation metric is sensible due to its deterministic nature, reducing ambiguity in retrieval evaluations.

**Other Comments Or Suggestions:**

1. Clarify runtime environment specifics (hardware configurations clearly in the main paper for reproducibility).

**Other Strengths And Weaknesses:**

Strengths:
1. Clear method contributions in using unsupervised methods effectively.
2. Robust and comprehensive experimental validation.
3. Effective demonstrations of practical applicability.

Weaknesses:
1. Limited analysis of robustness to dataset variability or extreme cases, especially in the Flickr dataset.

**Questions For Authors:**

1. How does the choice of embedding dimension affect SCENIR’s retrieval accuracy and computational cost?
2. Have you explored the robustness of SCENIR against noisy or incomplete scene graphs? If so, what were the findings?

**Relation To Broader Scientific Literature:**

SCENIR's contributions are clearly positioned against related literature. Its novelty primarily lies in extending unsupervised graph autoencoders for scene graph retrieval tasks and advocating GED over caption-based supervision. The authors have clearly discussed prior related works such as IRSGS and Graph Counterfactuals.

**Theoretical Claims:**

n/a

---

> ### Author Rebuttal · Authors · 2025-03-30
>
> We sincerely thank Reviewer kfau for their thorough feedback and for recognizing the validity of our evaluation methods, the soundness of our experiments, and the clarity of our presentation. We address their reported limitations and respond to their questions below.
>
> - Other comments and suggestions: Regarding the runtime environment specifics, we note that all experiments were conducted on an NVIDIA Tesla P100 GPU (line 269). Given the page limit constraints, we prioritized presenting our core contributions concisely. However, if the paper is accepted, we will include additional details in the camera-ready version to further enhance reproducibility. For reference, our setup includes: a single NVIDIA Tesla P100 GPU (16GB VRAM), a preprocessed scene graph dataset (~2GB), Python 3.10, PyTorch Geometric 2.4.0, PyTorch 2.2.0, and CUDA 11.8.
> - Questions For Authors - 1: Regarding the impact of embedding dimension on retrieval accuracy and computational cost, we would like to emphasize that these results stem from hyperparameter tuning, where we empirically determined the embedding dimension alongside other hyperparameters (e.g., loss terms, encoder architecture). To address the reviewer’s valid concern, we would like to emphasize that the significant computational efficiency demonstrated in Table 5 is primarily due to SCENIR’s unsupervised nature. Specifically, its linear training time stems from the fact that it does not regress on a pair’s similarity value, and its linear preprocessing time arises because we do not compute similarity values for every pair—an obligatory step in the other two frameworks. This distinction is further explained in Section 4.4 of the paper.
> - Questions For Authors - 2/W1: Regarding SCENIR’s robustness against noisy or incomplete scene graphs, we would like to clarify that the primary focus of our work is to evaluate and compare different retrieval frameworks rather than to conduct an in-depth robustness analysis for noisy or out-of-domain inputs. To ensure a high-quality benchmark, we use the PSG dataset, which significantly refines annotations from previous datasets (as mentioned in lines 254-258). Additionally, we apply preprocessing steps (detailed in Appendix A) to construct the final train/test sets. That being said, some level of noise remains inevitable. For example, this has been highlighted by the deliberate inclusion of Figure 11 (left). Specifically, it can be observed that background objects may still appear in the scene graph. However, SCENIR effectively identifies the primary semantic object of interest (e.g., “train”) despite the presence of surrounding noise, demonstrating its ability to focus on key scene elements.
>
> We once again thank the reviewer for their valuable insights and suggestions, which help strengthen the clarity and impact of our work.

---

### Official Review · Reviewer_s1fN · 2025-03-18

**Overall Recommendation:** 4

**Summary:**

This paper tackles the problem of image-to-image retrieval, focusing on improving the retrieval performance by emphasizing semantic content over low-level visual features. The authors argue that current models often rely on visual biases (e.g., colors, lighting conditions) rather than semantic contents/understanding. To avoid bypassing semantic information, the authors recast this task into scene graph retrieval and  propose SCIENIR, which is an unsupervised graph autoencoder framework for scene graph retrieval. The model architecture is essentially a branched VGAE, which is trained with a combination of losses such as the reconstruction loss, adversarial loss, and KL loss (against Gaussian).

In the experiments, this approach is evaluated in two data settings, with/without scene graph annotation. For the dataset without scene graph annotation, captions and scene graphs are automatically generated with existing tools (BLIP and PSGTR). Graph edit distance is used as the similarity score throughout the experiments. The quantitative results show that the proposed method outperforms various baselines ranging from existing GNN-based approaches to VLMs. Additionally, the authors provide qualitative analysis, counterfactual retrieval results, and speed analysis, indicating SCIENIR's effectiveness and efficiency.

**Claims And Evidence:**

In my opinion, the most important and interesting claim is that an unsupervised graph autoencoder can effectively retrieve images, even surpassing its supervised counterparts. The quantitative results clearly support that the proposed approach has advantages over the baselines. I feel the baselines look a bit old. I discuss this point in the “Methods And Evaluation Criteria” section. Also, the authors advocate that GED is a more reliable measure for scene graph similarity than caption-based approaches. Figure 2 and Figure 7 demonstrate disagreement of those approaches and support this claim. Overall, this paper clarifies its claims and provides support evidence to them properly.

**Essential References Not Discussed:**

N/A

**Experimental Designs Or Analyses:**

See my comments in the “Methods And Evaluation Criteria” section.

**Methods And Evaluation Criteria:**

The proposed framework and the motivations behind it sound reasonable. My only concern is the choice of baselines. As of now, there are many strong captioning models available (e.g., LLaVA). Also, it would be nice to make some connection with SOTA LLMs such as GPT and Gemini models. It would be a reasonable baseline for the idea of scene graph retrieval (e.g., generate detailed image descriptions for all images and retrieve them in the text space using any text encoder). The model sizes are too different, but it would be great if the proposed approach achieves comparable results.

**Other Comments Or Suggestions:**

N/A

**Other Strengths And Weaknesses:**

- Overall, this paper is well-written and easy to follow. Supplementary materials typically provide additional information to clarify ambiguous points in the main text.
- To clarify my position, my only concern is the choice of baselines. I had hoped that this paper would use more up-to-date ones.

**Questions For Authors:**

N/A

**Relation To Broader Scientific Literature:**

N/A

**Theoretical Claims:**

N/A

---

> ### Author Rebuttal · Authors · 2025-03-30
>
> We sincerely thank Reviewer s1fN for their thoughtful feedback and for the care they took in understanding our claims and evaluation methods. We greatly appreciate their recognition of the validity of our work and their positive assessment of its presentation. We would like to address their main concern to provide further clarity and ensure a shared understanding.
>
> We appreciate the reviewer’s suggestion and acknowledge the relevance of strong captioning models such as LLaVA and large language models (LLMs) like GPT and Gemini. However, to ensure a fair comparison, we adhere to replicating baseline models as proposed by their respective authors without extending them beyond their intended scope (e.g., substituting the captioner with an LLM-based alternative). More broadly, we believe that modifying the captioning module is slightly beyond the scope of our work. While a more advanced captioner could generate richer visual descriptions - potentially benefiting prior approaches that rely on caption-based supervision (e.g., IRSGS) - our core contribution is to move away from textual descriptions altogether. As discussed in the Introduction (lines 091-107) and further validated in Section 4.2 (Caption-driven disagreements), caption-based similarity models (e.g., SBERT variants) exhibit high variability, leading to inconsistent matching results depending on the specific SBERT model employed. Importantly, this issue persists regardless of improvements in caption quality.
>
> By instead leveraging scene graphs, we utilize Graph Edit Distance (GED) as a more deterministic and reliable supervision signal, reducing the ambiguity inherent in text-based representations. This is a key aspect of our second contribution (highlighted at the end of the Introduction) and is further supported by recent literature (Dimitriou et al., 2024). Additionally, our approach enables counterfactual retrieval via conceptual edits - a direction explored in recent explainability research (Dimitriou et al., 2024) - which would not be as naturally facilitated by caption-based methods.
>
> Moreover, our decision not to rely on LLM-based captioning models aligns with our goal of reducing computational burden while maintaining strong performance. As the reviewer has acknowledged, LLM-based methods tend to be significantly more resource-intensive. Our approach provides a more efficient alternative, demonstrating that high-quality retrieval can be achieved without the need for large-scale caption generation.
>
> We once again thank the reviewer for their insightful feedback and for encouraging further discussion on this aspect of our work.

---

### Decision · Program_Chairs · 2025-05-01

**Decision:**

Accept (poster)

**Comment:**

The paper proposes a scene graph-based retrieval framework emphasising semantic content over superficial image characteristics. The motivation is meaningful, and the proposed method is novel. All reviewers considered the paper as acceptance. Therefore, I recommend the acceptance.